# Innate Immune Gene Polymorphisms and COVID-19 Prognosis

**DOI:** 10.3390/v15091784

**Published:** 2023-08-22

**Authors:** Evangelos Bakaros, Ioanna Voulgaridi, Vassiliki Paliatsa, Nikolaos Gatselis, Georgios Germanidis, Evangelia Asvestopoulou, Stamatia Alexiou, Elli Botsfari, Vasiliki Lygoura, Olga Tsachouridou, Iordanis Mimtsoudis, Maria Tseroni, Styliani Sarrou, Varvara A. Mouchtouri, Katerina Dadouli, Fani Kalala, Simeon Metallidis, George Dalekos, Christos Hadjichristodoulou, Matthaios Speletas

**Affiliations:** 1Department of Immunology & Histocompatibility, Faculty of Medicine, University of Thessaly, 41500 Larissa, Greece; ebakaros@uth.gr (E.B.); vpaliatsa@uth.gr (V.P.); easvestop@uth.gr (E.A.); stamalexiou@uth.gr (S.A.); ebotsfari@uth.gr (E.B.); ssarrou@uth.gr (S.S.); fkalala@uth.gr (F.K.); 2Laboratory of Hygiene and Epidemiology, Faculty of Medicine, University of Thessaly, 41222 Larissa, Greece; ioavoulg@uth.gr (I.V.); mouchtourib@uth.gr (V.A.M.); adadouli@uth.gr (K.D.); xhatzi@uth.gr (C.H.); 3Department of Medicine and Research Laboratory of Internal Medicine, National Expertise Center of Greece in Autoimmune Liver Diseases, Full Member of the European Reference Network on Hepatological Diseases (ERN RARE-LIVER), General University Hospital of Larissa, 41110 Larissa, Greece; gatselis@me.com (N.G.); lvasiliki@med.uth.gr (V.L.); dalekos@uth.gr (G.D.); 4First Internal Medicine Department, Infectious Diseases Division, AHEPA Hospital, Medical School, Aristotle University of Thessaloniki, 54636 Thessaloniki, Greece; geogerm@auth.gr (G.G.); olgat_med@hotmail.com (O.T.); jordymim@hotmail.com (I.M.); metallidissimeon@yahoo.gr (S.M.); 5National Public Health Organization, 15123 Athens, Greece; mtseroni@nurs.uoa.gr

**Keywords:** COVID-19, IL18, TLR2, TLR4, CD14, CD40, CARD8, innate immunity, polymorphism, prognosis

## Abstract

COVID-19 is characterized by a heterogeneous clinical presentation and prognosis. Risk factors contributing to the development of severe disease include old age and the presence of comorbidities. However, the genetic background of the host has also been recognized as an important determinant of disease prognosis. Considering the pivotal role of innate immunity in the control of SARS-CoV-2 infection, we analyzed the possible contribution of several innate immune gene polymorphisms (including *TLR2*-rs5743708, *TLR4*-rs4986790, *TLR4*-rs4986791, *CD14*-rs2569190, *CARD8*-rs1834481, *IL18*-rs2043211, and *CD40*-rs1883832) in disease severity and prognosis. A total of 249 individuals were enrolled and further divided into five (5) groups, according to the clinical progression scale provided by the World Health Organization (WHO) (asymptomatic, mild, moderate, severe, and critical). We identified that elderly patients with obesity and/or diabetes mellitus were more susceptible to developing pneumonia and respiratory distress syndrome after SARS-CoV-2 infection, while the *IL18*-rs1834481 polymorphism was an independent risk factor for developing pneumonia. Moreover, individuals carrying either the *TLR2*-rs5743708 or the *TLR4*-rs4986791 polymorphisms exhibited a 3.6- and 2.5-fold increased probability for developing pneumonia and a more severe disease, respectively. Our data support the notion that the host’s genetic background can significantly affect COVID-19 clinical phenotype, also suggesting that the *IL18*-rs1834481, *TLR2*-rs5743708, and *TLR4*-rs4986791 polymorphisms may be used as molecular predictors of COVID-19 clinical phenotype.

## 1. Introduction

COVID-19 disease, which is caused by the emerging coronavirus SARS-CoV-2, is characterized by a heterogeneous and wide spectrum of clinical symptoms and prognoses. While most of infected individuals experience asymptomatic or mild disease, a considerable number of patients with SARS-CoV-2 infection can progress to a more severe and/or critical disease, accompanied by potentially life-threatening manifestations and complications [1]. Previous studies have already demonstrated that risk factors contributing to the development of severe COVID-19 include old age, low baseline ratio of arterial partial pressure of oxygen to fraction of inspired oxygen (pO_2_/FiO_2_), nationality, and the presence of comorbidities, such as obesity, diabetes mellitus, and hypertension [1,2,3,4,5,6].

Moreover, the host’s genetic background represents one of the most crucial parameters that modulate COVID-19 susceptibility and severity [7,8,9,10]. Several studies have already pointed out that variations in selected genes of the host play a pivotal role in disease progression, affecting the different COVID-19 clinical phenotypes and outcomes [7,8,9]. In this context, we and others have also analyzed functional innate immune gene polymorphisms (such as *MBL2*, *TLR7,* etc.), considering that the innate immune system plays a critical role in COVID-19 prognosis and outcome [3,11,12]. It has been established with certainty that innate immunity can determine early control of the SARS-CoV-2 infection, and its dysregulation may result in hyperinflammation events, which could lead to more severe disease and worse prognosis [1,13,14]. Thus, we could speculate that additional functional polymorphisms of the innate immune system may represent important risk factors for COVID-19 phenotype and outcome. 

In this study, we explored the contribution of additional known functional innate immune gene polymorphisms, namely, *TLR2*-rs5743708, *TLR4*-rs4986790, *TLR4*-rs4986791, *CD14*-rs2569190, *CARD8*-rs1834481, *IL18*-rs2043211, and *CD40*-rs1883832, in COVID-19 phenotype and prognosis since data on these polymorphisms are either limited or missing. 

## 2. Materials and Methods

### 2.1. Subjects

A total of 249 individuals (male/female: 172/77, median age: 38.0 years, 18.0–89.0) were enrolled in the study. Participants’ ethnicities included 166 patients of European origin (145 Greeks), 78 of Asian origin (62 Turks), and 5 of American origin (3 Americans and 2 Cubans); most patients of non-Greek origin were cruise ferry passengers and crew members who became infected with COVID-19 during March 2020 [15]. All patients were infected by the initial Wuhan-Hu-1 strain of SARS-CoV-2 between March and October 2020, as confirmed by RNA reverse transcriptase-polymerase chain reaction (RT-PCR) testing on nasopharyngeal swab specimens. 

Patients were further divided into five (5) groups, according to the World Health Organization (WHO) [16] guidelines and clinical progression scale as follows: (a) Asymptomatic (58, 23.3%). (b) Mild (88, 35.3%), including patients who had symptoms consistent with COVID-19, such as cough, chest pain, runny nose, anosmia, ageusia, digestive symptoms, headache or myalgia, but not exhibiting signs of viral pneumonia or hypoxia. (c) Moderate (26, 10.4%), including hospitalized symptomatic patients who had clinical pneumonia symptoms (fever, cough, dyspnea, tachypnea), but no signs of severe pneumonia, such as SpO_2_ < 90% on room air necessitating oxygen supplementation by mask or nasal prong. (d) Severe (34, 13.7%), including patients with clinical signs of pneumonia plus one of the following: breath rate > 30/min, severe respiratory distress or SpO_2_ < 90% on room air. (e) Critical (43, 17.3%), including patients with complications, such as respiratory failure, acute respiratory distress syndrome (ARDS), sepsis or septic shock, thromboembolism, and/or multi-organ failure, as well as patients who died. According to the aforementioned progression scale for COVID-19 severity, patients with pneumonia could be classified into either stage three which denoted moderate disease, or stage four indicating severe disease, depending on the patient’s clinical assessment. Similarly, patients experiencing respiratory distress could be categorized as either stage four, indicating severe disease or stage five, indicating critical disease, based on the severity of their medical condition, PaO_2_/FiO_2_ ratio, and the life support measures required during hospitalization. An overview of enrolled patients’ demographic and clinical characteristics is shown in Table 1. 

Written informed consent was obtained from all participants. The study was carried out in accordance with the principles of the Helsinki Declaration and was approved by the ethical committee of the Faculty of Medicine, University of Thessaly (No. 2115/22.04.2020).

### 2.2. Molecular Analyses

Genomic DNA was initially extracted from peripheral blood samples using the EXTRACTME GENOMIC DNA KIT (Blirt, Gdansk, Poland), according to the manufacturer’s instructions. DNA samples were stored at −20 °C until use. We further selected seven (7) functional single nucleotide polymorphisms (SNPs) in six (6) innate immune genes: *TLR2* (rs5743708), *TLR4* (rs4986790, rs4986791), *CD14* (rs2569190), *CARD8* (rs1834481), *IL18* (rs2043211), and *CD40* (rs1883832). SNPs were genotyped by PCR and restriction fragment length polymorphism (PCR-RLFP) assays, as follows: 100–200 ng of DNA was amplified in a 30 μL reaction mixture containing 62.5 μM of each deoxynucleoside triphosphate, 20 pmol of forward and reverse primers, 1.5 mM MgCl_2_, and 0.8 U of DFS-Taq DNA polymerase (Bioron GmbH, Romerberg, Germany) in a buffer provided by the manufacturer. Thereafter, PCR products were subjected to digestion with the appropriate restriction enzymes (New England Biolabs, Ipswich, UK) at 37 °C overnight; the digestion products were separated on 2–3% agarose gels and visualized by ethidium bromide staining. PCR and RFLP assays were conducted in an Applied Biosystems thermal cycler (Foster City, CA, USA). The sequences of the utilized primers, conditions of PCR reactions, and the restriction enzymes used for the detection of each SNP are shown in detail in Table 2. It is worth noting that the primers for the detection of *TLR2*-rs5743708, *TLR4*-rs4986790, *TLR4*-rs4986791, and *CD40*-rs1883832 polymorphisms were previously reported by us and others in the literature [17,18,19].

To confirm our PCR-RFLP results, randomly chosen wild-type, heterozygous, and homozygous PCR products were purified using a PCR purification kit (Qiagen, Crawley, UK), and subsequently sequenced using an ABI Prism 310 genetic analyzer (Applied Biosystems) and a BigDye Terminator DNA sequencing kit (Applied Biosystems). 

### 2.3. Statistical Analysis

Categorical variables are described with the use of frequency and relative frequency. Continuous variables are described with medians and interquartile range (IQR). Categorical data were analyzed with the use of Chi-square test after Yate’s correction or Fisher’s exact test. The analysis of continuous variables was conducted using the Mann−Whitney U test, as the assumption of normal distribution was violated. Data were checked for deviation from normal distribution using the Shapiro−Wilk normality test. Multivariate analysis was performed in the form of binary and ordinal logistic regression. A 5% significance level was set for all analyses. The analysis was carried out with SPSS version 29.0 (IBM Corp. Released 2021. IBM SPSS Statistics for Windows, Version 29.0. Armonk, NY, USA: IBM Corp).

## 3. Results

The distribution of patients’ genotype and allele frequencies regarding the analyzed SNPs are presented in Table 3. All genotype frequencies were in Hardy−Weinberg equilibrium (Table 3). We identified that the rs1834481 polymorphism significantly affected the clinical presentation of COVID-19, since rs1834481-G carriers displayed a significantly increased risk for developing pneumonia and severe disease (Table 4). 

In addition, we observed that carriers of TLR2-rs5743708 and TLR4-rs4986791 SNPs displayed a 3.7- and 2.2-fold increased risk for developing pneumonia, respectively, also affecting the risk for a more severe disease; however, the statistical differences were not marginally significant (Table 4). Thus, bearing in mind that both receptors exhibit common properties, they are the best characterized pattern recognition receptors (PRRs) and shape pathogen specific innate immunity through distinct ligand binding [20]. As detailed in Table 4, we further demonstrated that carriers of either TLR2 or TLR4 polymorphism, as a whole group, displayed a significant risk for developing pneumonia and more severe disease. 

Moreover, we identified that the CARD8-rs2043211 polymorphism affected the COVID-19 phenotype, providing a protective effect against severe disease (Table 4). Conversely, the other polymorphisms did not have any impact on COVID-19 severity (*p* > 0.05 in all cases; Table 4).

COVID-19 severity was also significantly affected by both age and the presence of comorbidities (Table 1). Indeed, elderly patients as well as patients with obesity, hypertension or diabetes mellitus were more likely to present severe COVID-19-associated clinical symptoms, such as pneumonia and respiratory distress syndrome (Table 1). 

Therefore, we performed multivariate analysis to identify risk factors affecting the clinical presentation and outcome of COVID-19, including all parameters that were ascertained in univariate analyses. Elderly patients with obesity and/or diabetes mellitus were more vulnerable to developing pneumonia and respiratory distress syndrome, following SARS-CoV-2 infection (Table 5). Moreover, individuals with a medical history of hematological and/or chronic immune-mediated diseases were more susceptible to developing pneumonia and severe disease, while patients with hypertension displayed an approximately 3-fold increased probability of developing more severe disease (Table 5). Additionally, we observed that the presence of the IL18-rs1834481 polymorphism was an independent risk factor for developing pneumonia, also displaying a 1.7-fold increased risk for more severe disease; although the latter was not marginally reached to statistical significance (Table 5). Finally, carriers of either TLR2-rs5743708 or TLR4-rs4986791 polymorphism also displayed a significantly increased risk for developing pneumonia and more severe disease (Table 5). An overview of the significant risk factors that contributed to the development of pneumonia in our cohort of COVID-19 patients is presented in Figure 1. 

## 4. Discussion

COVID-19 outcome is the result of a complex interplay between viral components and host proteins which are involved in the immune response to the virus [1,11,12]. However, soon after the COVID-19 outbreak, it became clear that the disease heterogeneity may be partially explained by the host’s genetic background [1,2,3,4,5,6,7,8,9,10,11,12]. In this context, the main objective of our study was to examine whether functional polymorphisms in genes coding for crucial mediators of innate immune response, namely, *TLR2*, *TLR4*, *CD14*, *CARD8*, *IL-18,* and *CD40*, can influence COVID-19 clinical severity in a multinational population of patients upon SARS-CoV-2 infection. 

One of the principal findings of our study was that the *IL18*-rs1834481 polymorphism appeared to have a strong impact on COVID-19 clinical outcome, since heterozygotes for the rs1834481 polymorphism were at a higher risk of developing pneumonia (both in univariate and multivariate analyses). IL-18 is a proinflammatory cytokine released downstream of the activation of innate immune pathways, resulting in the formation of inflammasomes; it has already been correlated with a poor COVID-19 prognosis, since increased IL-18 levels have been found in patients with severe disease [21,22].

The rs1834481 is an intronic SNP (located into intron 3) of the *IL18* gene [23]. Previous studies have demonstrated that along with other SNPs (namely, rs2115763 and rs5744256), rs1834481 is a functional polymorphism affecting *IL18* expression, and the presence of G alleles have significantly been associated with lower IL-18 serum levels [24,25]. The frequency of rs1834481-G allele is approximately 22.2% globally (https://www.ncbi.nlm.nih.gov/snp/rs1834481#frequency_tab, accessed on 12 July 2023), and its presence appears to protect carriers from coronary artery disease [26], non-small cell lung cancer [27] or human papilloma virus (HPV) persistence [28]. However, it has been associated with a poor prognosis of colon cancer [29]. According to our results, the rs1834481-G allele which has been associated with lower IL-18 levels after an immune stimulation, predisposes to more severe disease following SARS-CoV-2 infection. This may suggest that it could be used as a molecular predictor for COVID-19 severity and outcome. Considering that increased IL-18 levels were found in patients with a more severe COVID-19 phenotype, we could speculate that an inappropriate initial response against SARS-CoV-2 in rs1834481-G allele carriers may predispose to virus persistence, and subsequently to dysregulated immune responses.

Secreted IL-18 activates signaling pathways resulting in the production of IFNγ and other proinflammatory molecules, especially in CD8+ T cells, which represent crucial mediators of viral clearance [30,31]. In this context, reduced IL-18 production, due to rs1834481-G allele presence, may result in insufficient control of SARS-CoV-2 replication at the early stage of infection which could then lead to viral persistence. In the case that this immune checkpoint mechanism fails, repeated stimulation of IL-18-activated pathways could result in subsequent hyperinflammation events, and finally to more severe disease and poor outcome [14]. However, antiviral and inflammatory responses mediated by IL-18 are strongly affected by other cytokines and the immune cell milieu; therefore, further investigation is needed to completely unravel the role of IL-18 in COVID-19 [30].

To the best of our knowledge, this is the first study in the literature indicating a strong association between *IL18*-rs1834481 and clinical severity of COVID-19. After a broad literature review, we found only one additional study analyzing the contribution of rs1834481 in COVID-19 severity. Leal et al. analyzed a larger cohort of Brazilian patients infected by SARS-CoV-2, but they did not observe any significant association of rs1834481 with COVID-19 severity; however, they used a different methodological approach comparing only patients with severe and mild disease, while they did not use, at least, the WHO classification as in our study [32].

Additionally, we clearly demonstrated that carriers of either *TLR2*-rs5743708 or *TLR4*-rs4986791 polymorphisms, as a whole group, displayed a significant risk for pneumonia development and more severe disease (Table 4 and Table 5). The aforementioned polymorphisms have been associated with receptor hypo-responsiveness and susceptibility to several infectious diseases (reviewed by Schroder and Schumann) [33]. Accumulating evidence demonstrates that the viral S protein of SARS-CoV-2 may bind to and activate TLR4, while TLR2 was shown to interact with the envelope protein of the virus [34,35,36]. The interaction of E and S proteins with TLR2 and TLR4, respectively, induces intracellular signaling pathways that result in the transcriptional activation and expression of inflammatory genes, IFN genes, and interferon-stimulated genes (reviewed by Mantovani S., et al.) [37]. In the presence of *TLR2*-rs5743708 and/or *TLR4*-rs4986791 polymorphisms, reduced binding affinity for E and S proteins, respectively may explain the inadequate immune response against SARS-CoV-2 that fails to control infection, worsening the disease outcome. Therefore, our data further support the results of two previous studies from Egypt, where a positive correlation between the aforementioned polymorphisms and COVID-19 severity was observed [38,39], suggesting that both SNPs could be used as molecular predictors of COVID-19 prognosis and outcome.

Among the other innate immune gene polymorphisms analyzed, the *CARD8*-rs2043211 polymorphism was found to exhibit a statistically significant association. CARD8 negatively regulates inflammasome activation via inhibiting caspase 1 and NF-κB-mediated signaling pathways [40]. Presence of the polymorphism results in a substitution of phenylalanine by isoleucine at position 102 in the classical transcript variant 1 of CARD8 gene (NM_001184900.3), although it could lead to a termination codon in other splicing variants of the gene (transcript variants 12, 14, and 17–19; https://www.ncbi.nlm.nih.gov/snp/rs2043211#variant_details, accessed on 12 July 2023). Similar to the study conducted by de Sá et al. [41], in our cohort we observed a protective effect of *CARD8*-rs2043211 polymorphism in COVID-19 severity (Table 4); however, this finding was lost in multivariate analysis (Table 5).

CD14 acts as a coreceptor for both of the aforementioned TLRs, facilitating the binding of several ligands (pathogen-associated molecular patterns (PAMPS) and danger-associated molecular patterns (DAMPS)) to their receptors [42]. Although Pati et al. suggested that *CD14*-rs2569190 which is located in the promoter region of the *CD14* gene, may be linked to SARS-CoV-2-assocociated mortality in the European population [43], our findings do not support a positive association of this polymorphism with COVID-19 severity and outcome.

Finally, CD40 is a pivotal immune checkpoint molecule expressed by professional antigen-presenting cells, acting as a bridge between innate and adaptive immune responses [44]. To the best of our knowledge, this is the first study analyzing the contribution of the *CD40*-rs1883832 polymorphism with clinical presentation and severity of SARS-CoV-2 infection. Our group has recently reported that upon vaccination, this polymorphism affects the IgA antibody responses against SARS-CoV-2 [19], while it has also been correlated with a susceptibility to sepsis in a Chinese population [45]. However, we did not find a significant association between the *CD40*-rs1883832 polymorphism and the clinical course of COVID-19 in our study.

## 5. Conclusions

In conclusion, our study provides clear evidence that the host’s genetic background can significantly affect COVID-19 clinical presentation and severity, underlining the role of immunogenetics in COVID-19 prognosis. The identification of molecular markers, such as the *IL18*-rs1834481, *TLR2*-rs5743708, and *TLR4*-rs4986791 polymorphisms, and the incorporation of this knowledge into clinical practice may allow for better stratification of patients for an improved therapeutic intervention.

## Figures and Tables

**Figure 1 viruses-15-01784-f001:**
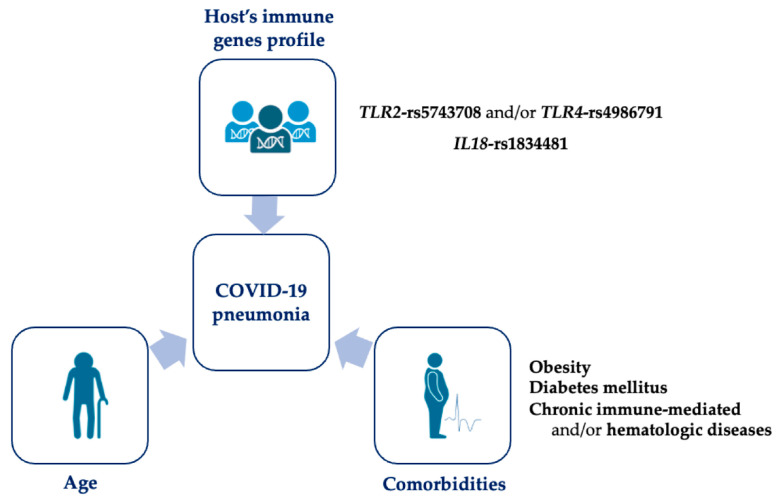
Risk factors that contributed to pneumonia development in our cohort of COVID-19 patients (Created with BioRender.com, accessed on 3 December 2022).

**Table 1 viruses-15-01784-t001:** Demographic and clinical characteristics of the study patients. Patients are divided into five (5) categories according to guidelines and the clinical progression scale provided by the World Health Organization (WHO), as detailed in the text.

	Total	Asymptomatic Disease	Mild Disease	Moderate Disease	Severe Disease	Critical Disease	*p* Value #
No	249	58	88	26	34	43	-
Sex (male/female)	172/77	40/18	60/28	16/10	27/7	29/14	0.648 (C)
Age	42.8 ± 18.5	35.3 ± 14.3	31.3 ± 11.6	48.2 ± 17.3	58.1 ± 14.8	60.8 ± 14.6	**<0.001** (K−W)
Comorbidities (n, %)	89, 35.7	6, 10.3	9, 10.2	15, 57.7	23, 67.6	36, 83.7	**<0.001** (C)
Obesity (n, %)	28, 11.2	0, 0.0	2, 2.3	4, 15.4	4, 11.8	18, 41.9	**<0.001** (C)
Hypertension (n, %)	45, 18.1	1, 1.7	2, 2.3	5, 19.2	12, 35.3	25, 58.1	**<0.001** (C)
Dyslipidemia (n, %)	17, 6.8	1, 1.7	0, 0.0	5, 19.2	4, 11.8	7, 16.3	**<0.001** (C)
Diabetes mellitus (n, %)	20, 8.0	0, 0.0	2, 2.3	1, 3.8	8, 23.5	9, 20.9	**<0.001** (C)
Thyroid disease (n, %)	10, 4.0	0, 0.0	3, 3.4	3, 11.5	1, 2.9	3, 7.0	0.116 (C)
CHD (n, %)	11, 4.4	0, 0.0	0, 0.0	1, 3.8	4, 11.8	6, 14.0	**<0.001** (C)
CVD (n, %)	8, 3.2	0, 0.0	0, 0.0	1, 3.8	0, 0.0	7, 16.3	**<0.001** (C)
CRD (n, %)	18, 7.2	1, 1.7	1, 1.1	0, 0.0	6, 17.6	10, 23.3	**<0.001** (C)
Chronic liver and/or renal disease (n, %)	6, 2.4	1, 1.7	0, 0.0	2, 7.7	1, 2.9	2, 4.7	0.176 (C)
Malignancies (n, %)	6, 2.4	0, 0.0	2, 2.3	0, 0.0	1, 2.9	3, 7.0	0.204 (C)
Chronic immune and/or hematologic diseases (n, %) ^	6, 2.4	1, 1.7	0, 0.0	0, 0.0	3, 8.8	2, 4.7	**0.044** (C)
Comorbidity-others (n, %) *	15, 6.0	3, 5.2	0, 0.0	1, 3.8	5, 14.7	6, 14.0	**0.004** (C)

Abbreviations: CHD, chronic heart disease (atrial fibrillation, heart failure, arrhythmias, prosthetic valve); CRD, chronic respiratory diseases (chronic obstructive pulmonary disease, asthma, sleep apnea); CVD, cerebrovascular disease (including chronic venous disease); SD, standard deviation. ^ Chronic immune and/or hematologic diseases included patients with systemic lupus erythematosus (1), autoinflammatory disease (1), IgA deficiency (1), beta-thalassemia (1), sickle-cell disease (1), splenectomy (unknown etiology) (1). * Comorbidity-others included patients with history of transplantation (liver or kidney), history of severe infections (meningitis, flu due to H1N1, recurrent urinary infections), history of major surgery (partial gastrectomy, nephrectomy, partial colectomy), dementia, depression. # Statistical analysis: C, Chi-square; K−W, Kruskal−Wallis test; M−W, Mann−Whitney U test.

**Table 2 viruses-15-01784-t002:** PCR-RFLP protocols for the detection of single nucleotide polymorphisms (SNP) of innate immune genes of the study.

Gene	SNP	Primer Sequences	PCR Conditions	PCR Product (bp)	Digestion Enzyme	Expected RFLP Fragments (bp) *
*TLR2*	rs5743708	F: 5′-TATGGTCCAGGAGCTGGAGA-3′	94 °C 2 min, 38 cycles (94 °C 30 s, 64 °C 30 s, 72 °C 45 s), 72 °C 5 min	430	SfcI	430 (wt/GG), 430 + 307 + 123 (heter/GA), 307 + 123 (homo/AA)
		R: 5′-TGACATAAAGATCCCAACTAGACAA-3′				
*TLR4*	rs4986790	F: 5′-GATTAGCATACTTAGACTACTACCTCCATG-3′	94 °C 2 min, 32 cycles (94 °C 30 s, 57 °C 30 s, 72 °C 60 s), 72 °C 5 min	249	NcoI	249 (wt/AA), 249 + 218 + 31 (heter/AG), 218 + 31 (homo/GG)
		R: 5′-GATCAACTTCTGAAAAAGCATTCCCAC-3′				
	rs4986791	F: 5′-GGTTGCTGTTCTCAAAGTGATTTTGGGAGAA-3′	94 °C 2 min, 32 cycles (94 °C 30 s, 57 °C 30 s, 72 °C 60 s), 72 °C 5 min	404	HinfI	404 (wt/CC), 404 + 375 + 29 (heter/CT), 375 + 29 (homo/TT)
		R: 5′-ACCTGAAGACTGGAGAGTGAGTTAAATGC-3′				
*CD14*	rs2569190	F: 5′-GCCAACAGATGAGGTTCACA-3′	94 °C 2 min, 30 cycles (94 °C 30 s, 60 °C 30 s, 72 °C 30 s), 72 °C 1 min	641	HaeIII	308 + 174 (wt/CC), 522 + 308 + 174 (heter/CG), 522 (homo/GG)
		R: 5′-GTTCGACCCCAAGACCCTAC-3′				
*CARD8*	rs2043211	F: 5′-AAGAGCAGCATATAGTGGGAAA-3′	94 °C 2 min, 30 cycles (94 °C 30 s, 57 °C 30 s, 72 °C 30 s), 72 °C 1 min	475	MboII	151 + 324 (wt/AA), 475 + 324 + 151 (heter/AT), 475 (homo/TT)
		R: 5′-TGAGTTCGATGAAAAACACCC-3′				
*IL18*	rs1834481	F: 5′-TTCTGTTGTAGGGGGAGCGT-3′	94 °C 2 min, 30 cycles (94 °C 30 s, 60 °C 30 s, 72 °C 30 s), 72 °C 1 min	522	HaeIII	308 + 174 (wt/CC), 522 + 308 + 174 (heter/CG), 522 (homo/GG)
		R: 5′-CATCAGGTTGCTTCCAGCCA-3′				
*CD40*	rs1883832	F: 5′-ATAGGTGGACCGCGATTGGT-3′	95 °C 10 min, 32 cycles (94 °C 30 s, 58 °C 30 s, 72 °C 30 s), 72 °C 10 min	216	NcoI	126 + 90 (wt/CC), 216 + 126 + 90 (heter/CT), 216 (homo/TT)
		R: 5′-TCCCAACTCCCGTCTGGT-3′				

Abbreviations: bp, base pairs; F, forward; RFLP, restriction fragment length polymorphism; R, reverse. * Within brackets we present the type of genotype (wt, wild-type; het, heterozygote; hom, homozygote) and the corresponding nucleotides of each SNP.

**Table 3 viruses-15-01784-t003:** Genotype and allele frequencies of analyzed single nucleotide polymorphisms (SNPs) of the study.

Gene	SNP	Genotype			Allele Frequency (n, %)		HWE
		WT (n, %)	Heter (n, %)	Homo (n, %)			
*TLR2*	rs5743708	239 (96.0)	10 (4.0)	0 (0.0)	G	A	0.75
	(g.25877G>A)				(488, 98.0)	(10, 2.0)	
*TLR4*	rs4986790	229 (92.0)	19 (7.6)	1 (0.4)	A	G	0.38
	(g.13622A>G)				(477, 95.8)	(21, 4.2)	
	rs4986791	230 (92.4)	18 (7.2)	1 (0.4)	C	T	0.33
	(g.13922C>T)				(478, 96.0)	(20, 4.0)	
*CD14*	rs2569190	61 (24.5)	123 (49.4)	65 (26.1)	T	C	0.85
	(g.5371T>C)				(245, 49.2)	(253, 50.8)	
*IL18*	rs1834481	187 (75.1)	54 (21.7)	8 (31.2)	C	G	0.11
	(g.16014C>G)				(428, 85.9)	(70, 14.1)	
*CARD8*	rs2043211	99 (37.8)	119 (47.8)	31 (12.4)	A	T	0.60
	(g.26821A>T)				(317, 63.7)	(181, 36.3)	
*CD40*	rs1883832	108 (43.4)	119 (47.8)	22 (8.8)	C	T	0.18
	(g.5077C>T)				(335, 67.3)	(163, 32.7)	

Abbreviations: Heter, heterozygous; Homo, homozygous; HWE, Hardy−Weiberg equilibrium; WT, wild-type.

**Table 4 viruses-15-01784-t004:** Innate immune gene single nucleotide polymorphisms (SNPs) and clinical phenotype of COVID-19.

Gene	SNP	Genotype (N, %) ^	Pneumonia (Yes vs. No)		Respiratory Distress (Yes vs. No)		Disease Severity (According to the WHO Classification) *	
			*p*	OR (95% CI)	*p*	OR (95% CI)	*p*	OR (95% CI)
*TLR2*	rs5743708	WT (239, 96.0) Heter + Homo (10, 4.0)	0.061	3.73 (0.94–14.78)	0.255	2.13 (0.58–7.81)	0.103	2.57 (0.83–8.03)
*TLR4*	rs4986790	WT (229, 92.0) Heter + Homo (20, 8.0)	0.329	1.75 (0.63–3.93)	0.789	0.75 (0.24–2.35)	0.305	1.53 (0.68–3.47)
	rs4986791	WT (230, 92.4) Heter + Homo (19, 7.6)	0.093	2.22 (0.86–5.73)	0.745	1.11 (0.38–3.22)	0.058	2.14 (0.92–4.94)
*CD14*	rs2569190	WT (65, 26.1) Heter + Homo (184, 73.9)	0.804	1.08 (0.60–1.92)	0.486	0.80 (0.42–1.51)	0.416	0.81 (0.49–1.34)
*IL18*	rs1834481	WT (187, 75.1) Heter + Homo (62, 24.9)	**0.012**	**2.09 (1.17–3.73)**	0.101	1.70 (0.90–3.20)	**0.008**	**2.01 (1.20–3.38)**
*CARD8*	rs2043211	WT (99, 39.8) Heter + Homo (150, 60.2)	0.220	0.72 (0.43–1.21)	0.084	0.72 (0.43–1.21)	**0.031**	**0.60 (0.38–0.96)**
*CD40*	rs1883832	WT (108, 43.4) Heter + Homo (141, 56.6)	0.665	1.14 (0.63–2.05)	0.442	1.22 (0.73–2.05)	0.368	1.23 (0.78–1.93)
*TLR2* or/and *TLR4*	rs5743708 or/and rs4986791	WT (223, 89.6) Heter + Homo(26, 10.4 ^#^)	**0.020**	**2.70 (1.17–6.22)**	0.432	1.43 (0.59–3.47)	**0.022**	**2.35 (1.13–4.88)**

Abbreviations: CO, confidence intervals; Heter, heterozygous; Homo, homozygous; OR, odds ratio; n.a., not-available (due to the low number of these groups of patients); WT, wild-type. The bold text highlights the statistically significant associations; the statistical analyses were performed using the Chi-square test for “pneumonia” and “respiratory distress” and the ordinal logistic regression analysis for “disease severity”. ^ WT genotype served as the reference for statistical analyses; * disease severity, as described in Material and Methods section. ^#^ Three (3) individuals were carriers of both polymorphisms (rs5743708 and rs4986791) in heterozygous state.

**Table 5 viruses-15-01784-t005:** Multivariate analysis of risk factors for COVID-19 phenotype and prognosis.

Parameter	Pneumonia		Respiratory Distress		Disease Severity (According to the WHO Classification) *	
	*p*	OR (95% CI)	*p*	OR (95% CI)	*p*	OR (95% CI)
Age	**<0.001**	**1.10 (1.06–1.13)**	**<0.001**	**1.06 (1.02–1.09)**	**<0.001**	**1.04 (1.02–1.06)**
Sex (female/male)	0.723	0.84 (0.31–2.24)	0.118	0.42 (0.14–1.24)	0.481	0.82 (0.47–1.43)
Obesity	**0.001**	**20.35 (3.46–119.68)**	**0.002**	**7.38 (2.13–25.61)**	**<0.001**	**8.18 (3.08–21.73)**
Hypertension	0.909	0.91 (0.19–4.34)	0.085	2.89 (0.86–9.65)	0.021	3.05 (1.18–7.87)
Diabetes mellitus	**0.029**	**12.15 (1.30–113.81)**	**0.013**	**6.59 (1.48–29.36)**	**0.039**	**3.32 (1.06–10.38)**
Dyslipidemia	0.195	4.46 (0.46–42.86)	0.813	1.21 (0.25–5.82)	0.801	1.16 (0.37–3.66)
Thyroid disease	0.323	2.85 (0.36–22.79)	0.489	2.05 (0.27–15.70)	0.220	2.29 (0.61–8.59)
CHD	0.999	-	0.305	0.37 (0.05–2.49)	0.801	0.82 (0.17–4.01)
CVD	0.999	-	0.172	5.61 (0.47–66.54)	0.055	9.26 (0.96–89.84)
CRD	0.775	1.31 (0.20–8.56)	0.307	2.05 (0.52–8.09)	0.058	3.01 (0.96–9.39)
Chronic liver and/or renal disease	0.754	1.54 (0.11–22.85)	0.749	0.67 (0.06–7.58)	0.917	0.91 (0.17–4.98)
Chronic immune and/or hematologic diseases	**0.002**	**57.89 (4.30–780.17)**	**0.216**	**4.63 (0.41–52.35)**	**0.004**	**11.92 (2.19–64.82)**
Malignancies	0.663	0.59 (0.06–6.28)	0.199	4.72 (0.44–50.53)	0.448	2.00 (0.33–12.05)
Comorbidity-others ^	0.897	1.13 (0.18–6.28)	0.892	1.12 (0.23–5.53)	0.466	1.55 (0.48–4.99)
*TLR2*-rs5743708 and/or *TLR4*-rs4986791	**0.033**	**3.62 (1.11–11.83)**	**0.897**	**1.09 (0.31–3.83)**	**0.026**	**2.54 (1.12–5.79)**
*IL18*-rs1834481	**0.026**	**2.88 (1.13–7.32)**	0.607	1.28 (0.50–3.26)	0.073	1.69 (0.95–2.99)
*CARD8*-rs2043211	0.751	1.15 (0.49–2.72)	0.210	0.57 (0.24–1.37)	0.204	0.72 (0.44–1.19)

Abbreviations: CHD, chronic heart disease (as in Table 1); CO, confidence intervals; CRD, chronic respiratory diseases (as in Table 1); CVD, cerebrovascular disease (as in Table 1); Heter, heterozygous; Homo, homozygous; WT, wild-type. The bold text highlights the statistically significant associations; the statistical analyses were performed using the binary logistic regression for “pneumonia” and “respiratory disstress” and the ordinal logistic regression analysis for “disease severity”. ^ As described in Table 1; * disease severity, as described in the Section 2.

## Data Availability

The data that support the findings of this study are available from the corresponding authors upon request.

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
