# Peer review of "Innate Immune Gene Polymorphisms and COVID-19 Prognosis"

_viruses, 2023, doi:10.3390/v15091784_

Round 1

Reviewer 1 Report

This article's text is well-written and informative. The tables are comprehensive and well-explained.  

Some suggestions are: 

  1. When discussing the IL18-rs1834481 polymorphism's impact on COVID-19 severity, you've done a great job explaining its potential biological significance. You could further expand on the implications of these findings. For example, how might the lower IL-18 levels associated with the rs1834481-G allele contribute to more severe disease? Are there potential therapeutic implications?

  2. In the discussion of TLR2-rs5743708 and TLR4-rs4986791 polymorphisms, you've mentioned their potential interaction with the viral S protein and their correlation with COVID-19 severity. To provide a comprehensive discussion, you could briefly explain how TLR2 and TLR4 are involved in the immune response to viral infections and how their hypo-responsiveness might impact the body's ability to control SARS-CoV-2.

Author Response

First of all, we would like to thank the reviewer for his/her supportive comments. Below we include a point-by-point response in his/her recommendations.

This article's text is well-written and informative. The tables are comprehensive and well-explained

Response: We would like to thank the reviewer for his/her supportive comments.

When discussing the IL18-rs1834481 polymorphism's impact on COVID-19 severity, you've done a great job explaining its potential biological significance. You could further expand on the implications of these findings. For example, how might the lower IL-18 levels associated with the rs1834481-G allele contribute to more severe disease? Are there potential therapeutic implications?

Response: According to the reviewer’s recommendation, we have provided the additional information on the association of IL-18 and the presence of rs1834481 polymorphism with COVID-19 severity in Discussion (lines 238-247). Accordingly, we have also provided the appropriate references (30-32).

In the discussion of TLR2-rs5743708 and TLR4-rs4986791 polymorphisms, you've mentioned their potential interaction with the viral S protein and their correlation with COVID-19 severity. To provide a comprehensive discussion, you could briefly explain how TLR2 and TLR4 are involved in the immune response to viral infections and how their hypo-responsiveness might impact the body's ability to control SARS-CoV-2.

Response: According to the reviewer’s recommendation, we have provided the additional information on the role of TLR2 and TLR4 in the immune responses to viral infections (especially against SARS-CoV-2) in Discussion (lines 270-276). Accordingly, we have also provided the appropriate reference (38).

Reviewer 2 Report

In this manuscript titled "Innate immune gene polymorphisms and COVID-19 prognosis", Bakaros et al addressed the role of innate immune system polymorphisms. They determined through regression analysis that at least the IL18-rs1834481, TLR2-rs5743708 and TLR4-rs4986791 polymorphisms are associated with the development of moderate/severe COVID 19.  

The topic is original because IL18-rs1834481 is reported in this work for the first time. It is relevant, because this polymorphism seems to be related to severe COVID-19 and the researchers propose it as a predictor of the severity of the disease. Although there have been several other genetic polymorphisms, this and other polymorphisms reported here needed to be evaluated.  

This work provides new insights into the immune polymorphisms related to COVID-19 that could help to understand the severity of the disease as well as to predict the impact of the infection on the health of patients.  

The protocol used in this work is based on the enzymatic digestion of amplicons obtained from the PCR method, which is suitable to test their hypothesis, although the sequencing of the genes reported here could be done to do a more robust work. The best control is the sequencing procedure.  

The conclusions are consistent with the results reported in the manuscript. The authors address all planning objectives in this work.  

Also, the references appropriate.  

The tables are complete, I just suggest adding supplementary information based on statistical tests. A cartoon based on the role of  the immune polymorphisms implicated in the development of COVID-19 will improve this manuscript.  

Therefore, the manuscript is suitable for publication

Author Response

First of all, we would like to thank the reviewer for his/her supportive comments. Below we include a point-by-point response in his/her recommendations.

In this manuscript titled "Innate immune gene polymorphisms and COVID-19 prognosis", Bakaros et al addressed the role of innate immune system polymorphisms. They determined through regression analysis that at least the IL18-rs1834481, TLR2-rs5743708 and TLR4-rs4986791 polymorphisms are associated with the development of moderate/severe COVID 19. The topic is original because IL18-rs1834481 is reported in this work for the first time. It is relevant, because this polymorphism seems to be related to severe COVID-19 and the researchers propose it as a predictor of the severity of the disease. Although there have been several other genetic polymorphisms, this and other polymorphisms reported here needed to be evaluated. This work provides new insights into the immune polymorphisms related to COVID-19 that could help to understand the severity of the disease as well as to predict the impact of the infection on the health of patients

Response: We would like to thank the reviewer for his/her supportive comments.

The protocol used in this work is based on the enzymatic digestion of amplicons obtained from the PCR method, which is suitable to test their hypothesis, although the sequencing of the genes reported here could be done to do a more robust work. The best control is the sequencing procedure

Response: We agree with the reviewer; however, as we have already presented in our manuscript (“Methodology”, lines 135-138), we have randomly chosen wild-type, heterogyzous, and homogyzous PCR products for sequencing analysis to validate our results. Thus, we consider that our approach, similarly with many manuscripts in the literature, was reproducible and extremely valuable.

The conclusions are consistent with the results reported in the manuscript. The authors address all planning objectives in this work. Also, the references appropriate

Response: We would like to thank the reviewer for his/her supportive comments.

The tables are complete, I just suggest adding supplementary information based on statistical tests

Response: According to the reviewer’s recommendation, we have added the information of the performed statistical analyses in Tables 4 and 5.

A cartoon based on the role of the immune polymorphisms implicated in the development of COVID-19 will improve this manuscript

Response: We agree with reviewer’s comment; thus, we have added an additional Figure (Figure 1), according to his/her recommendation.

Therefore, the manuscript is suitable for publication

Response: We would like to thank the reviewer for his/her recommendation.